# Prevalence of Comorbidities in Individuals Diagnosed and Undiagnosed with Alzheimer’s Disease in León, Spain and a Proposal for Contingency Procedures to Follow in the Case of Emergencies Involving People with Alzheimer’s Disease

**DOI:** 10.3390/ijerph17103398

**Published:** 2020-05-13

**Authors:** Macrina Tortajada-Soler, Leticia Sánchez-Valdeón, Marta Blanco-Nistal, José Alberto Benítez-Andrades, Cristina Liébana-Presa, Enrique Bayón-Darkistade

**Affiliations:** 1Facultad de Ciencias de la Salud, Campus de Vegazana, Universidad de León, s/n, C.P. 24071 León, Spain; macrina-96@hotmail.com; 2SALBIS Research Group, Facultad de Ciencias de la Salud, Campus de Ponferrada, Universidad de León, Avda/ Astorga s/n, C.P. 24402 Ponferrada (León), Spain; cristina.liebana@unileon.es (C.L.-P.); jebayd@unileon.es (E.B.-D.); 3Complejo Asistencial Universitario de León, C/ Altos de nava s/n, C.P. 24001 León, Spain; mblancon@saludcastillayleon.es; 4SALBIS Research Group, Department of Electric, Systems and Automatics Engineering, University of León, s/n, 24071 León, Spain; jbena@unileon.es

**Keywords:** Alzheimer’s disease, comorbidity, older adults, elderly

## Abstract

*Background*: Alzheimer’s disease (AD) which is the most common type of dementia is characterized by mental or cognitive disorders. People suffering with this condition find it inherently difficult to communicate and describe symptoms. As a consequence, both detection and treatment of comorbidities associated with Alzheimer’s disease are substantially impaired. Equally, action protocols in the case of emergencies must be clearly formulated and stated. *Methods:* We performed a bibliography search followed by an observational and cross-sectional study involving a thorough review of medical records. A group of AD patients was compared with a control group. Each group consisted of 100 people and were all León residents aged ≥65 years. *Results:* The following comorbidities were found to be associated with AD: cataracts, urinary incontinence, osteoarthritis, hearing loss, osteoporosis, and personality disorders. The most frequent comorbidities in the control group were the following: eye strain, stroke, vertigo, as well as circulatory and respiratory disorders. Comorbidities with a similar incidence in both groups included type 2 diabetes mellitus, glaucoma, depression, obesity, arthritis, and anxiety. We also reviewed emergency procedures employed in the case of an emergency involving an AD patient. *Conclusions:* Some comorbidities were present in both the AD and control groups, while others were found in the AD group and not in the control group, and vice versa.

## 1. Introduction

A general increase in life expectancy has caused an aging population and a resulting rise in the incidence of diseases that were less prevalent a few years ago such as neurodegenerative disorders [1]. At present, Alzheimer’s disease (AD) is the most common type of dementia [2,3,4,5,6,7,8,9,10,11,12,13,14,15,16,17]. It was first described in 1907 [9] by the German physician Alois Alzheimer [1], who diagnosed it in a 51-year-old woman. It was described as a disease characterized by an impaired memory, disorientation, and hallucinations leading to death [1]. Currently, AD is considered to be a neurodegenerative disease [1,9,11,12,13,15,17,18,19,20] that is progressive [4,11,12,17,18,19,20] and results in mental or cognitive dysfunctions [1,5,11,17].

AD has become a major world health problem affecting a continuously increasing number of people. In Spain, 500,000–800,000 people suffer from AD, a number expected to double by 2050 [2,3]. More specifically, it is calculated that 10% of the population aged ≥65 years and 50% of that ≥85 years will suffer from AD [18]. Aging, therefore, greatly increases the risk of AD [6,9,21], which by now has become a social and public health issue [1].

The main symptom of AD is the loss of episodic memory [4,9,10,14]. This is accompanied by other characteristic “warning signs” [4,6,9,20,22] namely:Failures or memory loss, which make everyday activities difficult;Difficulty to face and/or solve problems;Disorientation;Difficulty understanding visual images and spoken language;Problems with oral and/or written language;Placing objects out of place;Diminished and absent capacity of judgment;Loss of initiative;Personality changes, including apathy and depression;Higher anxiety levels, restlessness, and sleep disorders;Development of a state of increased dependency.

### 1.1. Pathophysiology of AD

The pathophysiology of AD is characterized by the occurrence of neurofibrillary tangles and neuritic plaques [2,3,5,10,12,17]. Several theories try to explain its onset [2,3,4,5,7,8,12,16,23,24] as follows:**Amyloid theory** The essential element of extracellular deposits is the protein *β*-amyloid, which forms fibrils that aggregate and cause the development of diffuse and neuritic plaques. The *β*-amyloid protein is produced by an abnormal cleavage of the amyloid precursor protein (APP). Normally, the product of secretase *α* action is a soluble peptide that can be easily removed from the body. In AD, the cleavage is performed by *β*- and *γ*-secretases producing insoluble peptides that are removed from neurons. Microglial cells unsuccessfully attempt their removal, and this results in inflammation and nerve damage.**Tau protein theory** The tau protein is the main component of intracellular deposits in neurons. It is a microtubule-associated protein, with microtubules being cytoplasmic structures involved in the assembly and function of the cytoskeletal network of cells including neurons. Tau acts as a microtubule stabilizer. In AD, Tau hyper-phosphorylation prevents its binding to tubulin and results in autoaggregation and formation of neurotoxic intraneuronal precipitates.**Cholinergic theory** A decrease in the levels of the neurotransmitter acetylcholine in patients with AD causes a diminished performance of neural connections.

In addition to the three theories mentioned above, several other hypotheses have attempted to explain the etiology of AD, such as oxidative stress and glutamate-mediated excitotoxicity [7,12].

### 1.2. Risk and Protective Factors

AD is associated with a series of risk and protective factors. A risk factor is understood as one that increases the probability that an individual will develop a health problem or disease; while a protective factor is one that reduces such probability. We present a list of such factors that are associated with AD as follows:

**Risk factors** [4,19,21,23,25,26,27,28,29,30,31]Non-modifiable factors include age, gender, genetics-related (karyotype alterations, gene mutations, etc.), parental education, family background;Modifiable factors include low educational or socioeconomic level, obesity, type II diabetes mellitus, cardiovascular diseases (hypertension, atherosclerosis, heart disease, atrial fibrillation, and dyslipidemia), smoking, stroke, depression, alcohol abuse, and pneumonia;Environmental factors [24] include aluminum, pesticides, pollution.

**Protective factors** [21] physical, educational, intellectual and social activities, moderate consumption of alcoholic beverages, and a Mediterranean diet.

Establishing which factors are protective or risk-linked for AD patients is made difficult by these patients’ inherent inability to communicate consistently. One way to approach this problem is by detecting comorbidities associated with AD and developing possible action protocols to be employed in emergency cases. The present study compared the comorbidities in a population of individuals aged ≥65 years and diagnosed with AD with those in an undiagnosed (control) population of similar characteristics, in the city of León, during 2019. In brief, the objectives of this work are as follows:To compare the sociodemographic characteristics of the individuals in the two populations under scrutiny;To establish and compare the comorbidities associated with the individuals of each of the two populations;To know, if any, the protocols of action employed by the Alzheimer Center of León or the León University Hospital in emergency situations concerning people with AD.

The cognitive or mental impairment that people with Alzheimer’s disease present increases the difficulty they have in expressing themselves and manifesting their symptoms. Therefore, there is a need to study the comorbidities associated with Alzheimer’s, and to examine possible protocols for action in the case of an emergency with these patients who have difficulty with expression and communication. This proposal for action protocols would facilitate emergency situations (such as triage in the emergency department) for these patients with other people, despite their difficulty with expression and communication.

## 2. Materials and Methods

### 2.1. Population Study

We performed an observational and cross-sectional study of the medical records of the populations under comparison. This involved a preliminary search strategy from primary and secondary bibliographic sources. A total of 200 individuals were analyzed, 100 from each of the 2 populations. 

A significance study was conducted using GPower software to estimate the sample size [32], then, two groups were established, i.e., control and AD, each consisting of 100 subjects, which provided a confidence level of 90%, on the basis of a total population with AD estimated in 7000 individuals, in the León region of Spain [33]. The Alzheimer’s Center León register contains 370 patients, with a proportion of them diagnosed with dementias other than Alzheimer’s (fluctuating percentage of about 20% diagnosed with other types of dementia, primarily frontotemporal dementia, Parkinson’s, and Lewy body dementia).

### 2.2. Literature Search

The bibliographic search connected with the present study included the PUBMED, WOS, and CUIDEN PLUS databases. A number of inclusion criteria were considered as follows:Publications from 2014 to 2019;Publications in English or Spanish;Data concerning the adult population aged ≥65 years diagnosed with AD;Both primary (original articles) and secondary (systematic reviews) sources;Full texts accessible through the University of León Library;Keywords employed (combined with Boolean operators AND and OR), in English included Alzheimer’s disease, comorbidity, elderly, and aged and, in Spanish, “Enfermedad de Alzheimer”, “comorbilidad”, “adulto mayor”, and “anciano”.

### 2.3. Data Collection

The current study was complied with the rules of the Helsinki Declaration of 1975. It was approved by the Ethics Committee of León University Hospital according to Resolution #1929 of 26 February 2019.

The criteria for inclusion in the AD study group were:Alzheimer’s disease diagnosis (this diagnosis has been previously identified by the Psychiatric Service of the León University Hospital);Age ≥65 years;Residency in the city of León, Spain;

The selection of subjects meeting these criteria was random. 

The criteria for inclusion in the control group were:Not diagnosed with AD;Age ≥65 years;Residency in the city of León, Spain.

#### 2.3.1. Alzheimer’s Disease (AD) Group

Data were obtained from the Alzheimer’s Comprehensive Care Center of León of the León Alzheimer Center. Each subject was identified by a code number to protect anonymity and confidentiality. This procedure was approved by the León Alzheimer Center and supported by signed agreements. Medical records were reviewed at random, reached the maximum number of individuals possible from the total number of people registered at the center, and met the inclusion criteria.

#### 2.3.2. Control Group

Data corresponding to the undiagnosed population were collected from the León University Hospital.

Subjects who were undiagnosed in the general population with Alzheimer’s disease, were randomly selected among those attending the Emergency Department of León University Hospital, for a few days, using the Gacela^®^ computer program (Oesía Group, Madrid, Spain), considering the following characteristics: age, sex, reason for admission, and medical history number. We used the Jimena^®^ software (Jimena software, Junta de Castilla y León, León; Spain) to review and collect data, which included the pathologies suffered and the list of drugs taken by each individual. Each subject was identified by a code number to protect anonymity and confidentiality. This procedure was approved by the León Alzheimer Center and supported by signed agreements.

#### 2.3.3. Data Processing

Data were stored and analyzed using an Office Excel^®^ spreadsheet processor (2019 version) (Microsoft Corporation, Redmond, Washington, USA). The data analyzed for both groups included age, sex, comorbidities, and number of drugs taken. It is important to mention that diagnoses of comorbidities were uniformly identified by both primary care and continuing care physicians. They were entered into an SPSS version 24 computer statistical program (IBM corp., New York, USA), which was followed by analysis of the variables for each of the 2 populations to be compared with each other.

Additionally, action protocols to be used in the case of an emergency involving a patient with Alzheimer’s disease were obtained. 

### 2.4. Significance Studies

Data significance level was calculated with Pearson’s Chi-square, and a value of *p* ≤ 0.1 was considered to be satisfactory. As mentioned in Section 2.1., the 2 populations of 100 individual analyzed, in the present study, provided a confidence level of 90% and *p* = 0.1.

## 3. Results

### 3.1. Sociodemographics

#### 3.1.1. Gender Distribution

The entire sample of 200 individuals comprising both the control and AD groups consisted of 61% women and 39% men. Figure 1 shows that the control group was composed of 47% males and 53% females; whereas the AD group consisted of 31% males and 69% females.

#### 3.1.2. Age Distribution

Figure 2 shows the comparative age analysis of the AD and control groups. This cohort study indicates that the number of subjects within the age interval 76–85 years is the largest in both groups. The age distribution of the AD and control groups clearly differs. The AD population shows an uneven pattern, with the >85 years group placed second after 76–85 years and a clearly smaller 66–75-year-old group. Instead, the distribution is rather symmetrical in the control population. In brief, the age interval in the AD group is shifted to older ages as compared with the control population.

### 3.2. Pathologies

The comparison of the AD group with the control group and the analysis of the pathologies observed in the populations under study shows three clear age range groups according to their higher, similar, or lower incidence of pathologies.

#### 3.2.1. Pathologies with a Higher Incidence in the AD Group

Table 1 shows the percentages of individuals with comorbidities more abundant in the AD group than in the controls. The pathologies in question are cataracts, urinary incontinence, vitamin D deficiency, osteoarthritis, hearing loss, osteoporosis, and personality disorders.

Cataracts, urinary incontinence, and vitamin D deficiency affect 21%, 38%, and 11% of the individuals in the AD group, respectively, doubling the values observed in the control group, which are 12%, 16% and 5%, respectively. The difference is even more pronounced for osteoarthritis, which was present in 26% of the individuals in the AD group as compared with 9% in the control group. Strikingly, only 1% of individuals (one subject) was affected by hypoacusis, osteoporosis, or personality disorder in the control group, while the values were 13%, 20%, and 12% in the AD group, respectively.

In particular, emphasis is placed on the relationship between osteoporosis and gender.

Figure 3 shows the difference in prevalence of osteoporosis between men and women in the control and AD groups. As indicated in Table 1, 1% of women in the control group have osteoporosis while there are no cases of men. In the AD group, a total of 17% of women and 3% of men have osteoporosis.

Table 2 indicates the Pearson’s Chi-square values and significance levels for the comorbidities shown in Table 1.

The significance levels of cataracts, urinary incontinence, osteoarthritis, hypoacusis, osteoporosis, and personality disorders were 0.086, <0.00, 0.002, 0.001, <0.00, and 0.002, respectively. These values were lower than the p-value of 0.1. Instead, vitamin D deficiency showed a significance level value of 0.118, i.e., higher than the *p*-value. Therefore, we can say that the differences observed for the conditions listed on Table 2 are significant, except for vitamin D deficiency.

#### 3.2.2. Pathologies with a Lower Incidence in the AD Group

Table 3 shows the percentages of individuals with comorbidities less abundant in the AD group as compared with the control group. The comorbidities which were analyzed included eye strain (increase in intraocular pressure), stroke, vertigo, hyperuricemia, circulatory insufficiency, atrial fibrillation, and respiratory insufficiency.

The AD group shows 2%, 1%, and 2% of individuals affected by eye strain, stroke, and vertigo, respectively; whereas the values in the control group are higher, i.e., 5%, 11%, and 14%, respectively.

The percentages of control individuals affected by hyperuricemia (increased uric acid in the blood), circulatory failure, and atrial fibrillation are 14%, 34%, and 11%, respectively. In the AD group, those percentages are 9%, 21%, and 7%, respectively; all of them lower than in the undiagnosed population. The percentage of individuals with respiratory failure is 25% in the control and 4% in the AD group, indicating an incidence five times higher in the control group.

Table 4 shows the Pearson’s Chi-square values corresponding to the comorbidities listed on Table 3.

The significance levels of the comorbidities eye strain, stroke, dizziness, circulatory insufficiency, and respiratory insufficiency are all below the *p*-value of 0.10. In contrast, the comorbidities hyperuricemia and atrial fibrillation have a significance level of 0.268 and 0.323, respectively, i.e., higher than the *p*-value of 0.10. Thus, we can say that the differences observed for the conditions listed on Table 4 are significant except for hyperuricemia and atrial fibrillation.

#### 3.2.3. Pathologies with a Similar Incidence in both Populations

Table 5 shows the comorbidities that do not show significant differences based on a comparison of the control and AD populations.

The proportions of individuals affected by type 2 diabetes mellitus, glaucoma, depression, obesity, arthritis, anxiety, and heart disease in the AD group are 19%, 6%, 27%, 7%, 8%, 14%, and 31%, respectively. In the control group, these values are 20%, 6%, 26%, 7%, 9%, 12%, and 26%, respectively. The figures show a similar incidence of these comorbidities in both the AD and control groups. The comorbidities arterial hypertension and dyslipidemia present a relatively high incidence in both the control and AD groups; with values, in the AD group, of 51% and 45%, respectively and, in the control group, the corresponding percentages are 64% and 39%, respectively.

Table 6 shows the significance levels of the comorbidities type 2 diabetes mellitus, glaucoma, depression, obesity, arthritis, dyslipidemia, anxiety, and heart disease are all well above the *p*-value of 0.10. Instead, hypertension shows a significance level of 0.063, which is lower than the *p*-value of 0.10. Thus, we can say that only the differences observed for hypertension are significant.

### 3.3. Medication

The number of medications taken by individuals from both study populations shows an average number of seven for each group, with the most abundant range being 6–10 medications.

Figure 4 shows that individuals in the control group took a larger number of drugs as compared with the AD group, with no one in the AD group taking >15 drugs. Nonetheless, the average number of drugs taken by each individual is the same (*n* = 7) in both groups.

## 4. Operating Procedures in the Case of Emergencies

This section outlines the analyses of possible courses of action in the case of emergencies involving AD patients. We re-examined the standard operating procedures employed in the two centers from which the data presented here were collected, namely the Alzheimer’s Comprehensive Care Center and the León University Hospital (León, Spain) (Table 7).

The protocol for the Alzheimer’s Comprehensive Care Center regarding patients with Alzheimer’s disease is focused on the prevention of falls and an action plan should a fall occur; whereas the protocol for the León University Hospital refers to the activities the nursing staff should perform according to the different nursing diagnosis lists of the NANDA (North American Nursing Diagnosis Association), for individuals with AD. The main difference between these two action protocols is that while that for the León University Hospital is exclusively directed to the nursing staff, the Alzheimer’s Comprehensive Care Center’s protocol is aimed at all its workers. Each of these approaches has different characteristics. A protocol of action, exclusively in the hands of the nursing staff, is based on their qualifications and competence to handle AD patients, and is standardized and excludes individual initiatives, thus, eliminating additional variables. An action plan aimed at all the workers dealing with AD patients has to be centered on the needs of such patients and must contemplate the fact that less qualified workers should seek professional advice whenever in doubt or when faced with unexpected events.

Fall prevention can require the physical restraint of AD patients, which must only be performed by medical order. In this respect, both centers have a specific protocol. When restraint requires the use of straps, the patient’s skin integrity is first assessed. Importantly, the subjects in question and their relatives must be informed. Furthermore, the Alzheimer’s Comprehensive Care Center’s protocol contains a scale of evaluation of fall risks according to the J.H. Downton scale [34]. This evaluation has been applied to all the individuals in this center, who are professionally supervised depending on their individual risks. The reliability of the risk scale must be regularly reassessed. The León University Hospital’s protocol does not use any scale to assess the risk of falls and simply considers the nature of the mental state alteration of each AD patient, for example, dementia, delirium, etc.

The Alzheimer Comprehensive Care Center also considers some intrinsic and extrinsic factors that affect fall risks. Classical intrinsic factors are age, medicines taken or associated comorbidities; while extrinsic factors are of an environmental nature, for example, inappropriate floor surfaces, lack of appropriate equipment, among others.

### Procedures in the Event of a Fall

Intervention in the case of a fall is protocolized in the Alzheimer Comprehensive Care Center but not in the León University Hospital.

The first step of the intervention is to help the person who fell, as well as reassure other individuals in the vicinity who witnessed the event. The latter is essential since AD patients are particularly sensitive to traumatic situations even if not personally involved, and their behavior can be altered. The second step is a professionally conducted evaluation of the fallen patient’s condition, with the help of an emergency team if necessary. It is extremely important to examine in detail why the fall occurred, in order to detect its possible causes, and therefore prevent other falls. It is equally important to consider the associated comorbidities of the patient, which may impinge on the action protocol to be used.

## 5. Discussion

The purpose of this work was to study the prevalence of comorbidities in an AD population as compared with a control population. First, we confirmed that the probability of developing AD is associated with an older population and is more frequent in females, in agreement with previous studies [26]. Presently, there is a debate as to whether AD is more prevalent among women due to genetic reasons or as a result of their longer life expectancy, which would make them more susceptible during later years.

Østergaard et al. (2015) [19] and Gallego and Guerrero (2017) [21] proposed that certain cardiovascular factors could facilitate the appearance of Alzheimer’s disease. Among them, they mentioned high blood pressure, type 2 diabetes mellitus, heart disease, dyslipidemia, and obesity. This view was shared by Dugger et al. [25]. Our results do not fully support these suggestions. In fact, we detected a lower incidence of arterial hypertension in the AD group as compared with the control group, which was significant (Table 5 and Table 6). No differences were observed between the AD group and the control group regarding the incidence of type 2 diabetes mellitus and obesity (Table 5). On the contrary, the incidence values for dyslipidemia and cardiopathy, although non-significant (*p* ≥ 0.1), indicate a higher prevalence in the AD group.

In our study, the incidence of glaucoma, depression, anxiety, and arthritis was similar, though not significant, in both the AD and control populations. Xu et al. [35] claimed, in 2019, that the correlation between glaucoma and Alzheimer’s disease was due to an enhanced susceptibility of AD patients to glaucoma. Depression and anxiety were described as AD predisposing factors by Ehrenberg et al., in 2019 [36]. Kao et al. [37] proposed that the correlation between arthritis and AD was actually an inverse relationship. According to our observations, there is a higher incidence of a history of stroke in the control group as compared with the AD group. The difference was significant and disagrees with previous studies by Nucera and Hachinski [27] and Hachinski [38], published in 2018, which showed that a previous history of stroke predisposed AD.

The comorbidities that we observed to be less prevalent in the AD group are ocular tension, vertigo, hyperuricemia, circulatory insufficiency, atrial fibrillation, and respiratory insufficiency. They are all significant except for hyperuricemia and atrial fibrillation. Lu et al. [39] reported, in 2017, a lack of a clear relationship between hyperuricemia and AD. No clear link between atrial fibrillation and AD was found by Ihara et al., in 2018 [40], while a positive correlation with AD was observed for vascular dementia.

The following comorbidities were found to be much more prevalent within the AD group as compared with the control group: cataracts, urinary incontinence, vitamin D deficiency, osteoarthritis, hearing loss, osteoporosis, and personality disorders. With the exception of vitamin D deficiency, their incidence values were all significant. In particular, in Figure 3 it is possible to see that there is a higher prevalence of osteoporosis in women than in men. This fact causes us to consider whether this comorbidity is associated with sex rather than a diagnosis of Alzheimer’s disease. For this reason, it is necessary to go deeper into the relationship between osteoporosis and Alzheimer’s disease and to see the factors that cause this comorbidity in patients with Alzheimer’s. The vitamin D deficiency results, though not significant, agree with those of Annweiler et al. [41] and Chen et al. [42]. The latter also found, similar to our findings, that osteoporosis appears to be associated with Alzheimer’s disease. Similarly, Lee et al. [43], Swords et al. [44], and Rouch et al. [45] reported the association of AD with urinary incontinence, hearing loss, and personality disorders. The fact that the AD group consists of more women (69%) than the control group (53%) may explain the higher incidence of osteoarthritis for the AD group [46]. Similarly, the fact that it is composed of an older population could also explain that this group of subjects has a greater number of associated comorbidities, not incidents in the same way as for the control group [46].

Altogether, our results show a satisfactory number of coincidences, as well as some discrepancies with those from other investigations. The discrepancies found could be due, among other possible aspects, to the difference between the sample size of our study with that of the other investigations or to the subjects chosen at random in one study or another.

## 6. Future Research

The observations reported, here, encourage further studies. First, an extension of the present research involving a greater number of patients and controls would be advisable. Secondly, longitudinal follow-up studies would allow the analysis of the existing and developing comorbidities of AD patients over long periods of time. As far as the Alzheimer’s-free population is concerned, longitudinal studies would enable the detection of people that develop AD related to ageing. It would be possible to document whether the comorbidities they already suffered from were affected or not after the onset of AD, or whether the comorbidities were in any way related to the onset of AD itself. Such retrospective studies would obviously require a thorough analysis of medical records over long periods of time.

## Figures and Tables

**Figure 1 ijerph-17-03398-f001:**
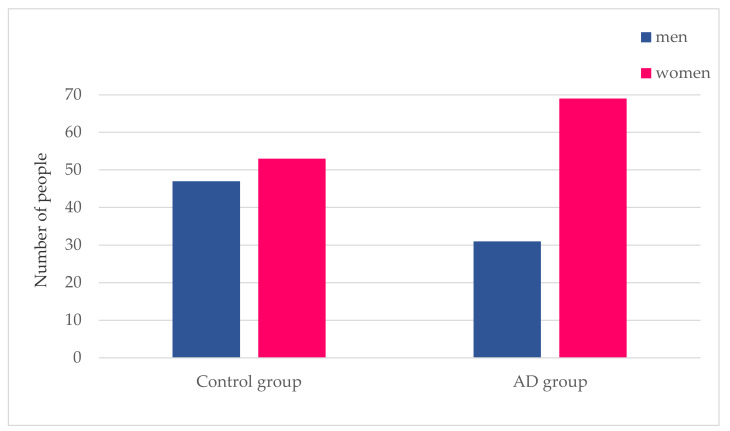
Alzheimer’s disease and gender. This figure shows the number of men and women that make up both the control group and the Alzheimer’s disease (AD) group.

**Figure 2 ijerph-17-03398-f002:**
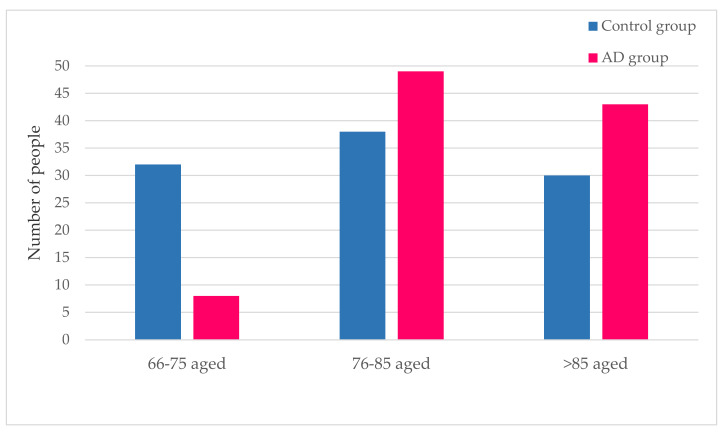
Age range distributions of the AD and controls. This figure shows the distribution of both the control group and the AD group, into three age range groups.

**Figure 3 ijerph-17-03398-f003:**
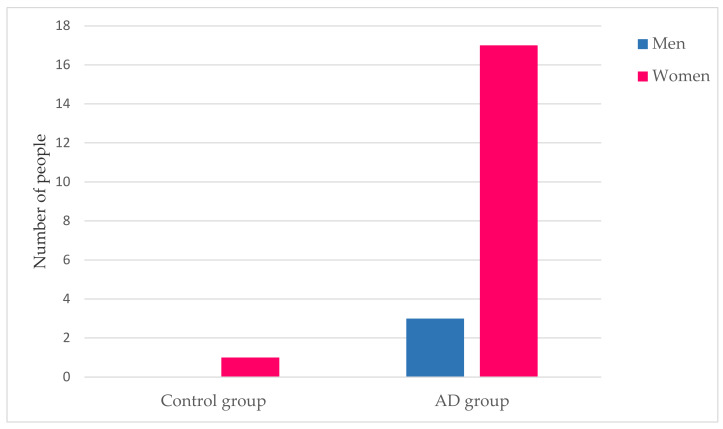
Osteoporosis’ disease and gender. This figure shows the percentage of men and women who have osteoporosis, differentiated into the two groups under study.

**Figure 4 ijerph-17-03398-f004:**
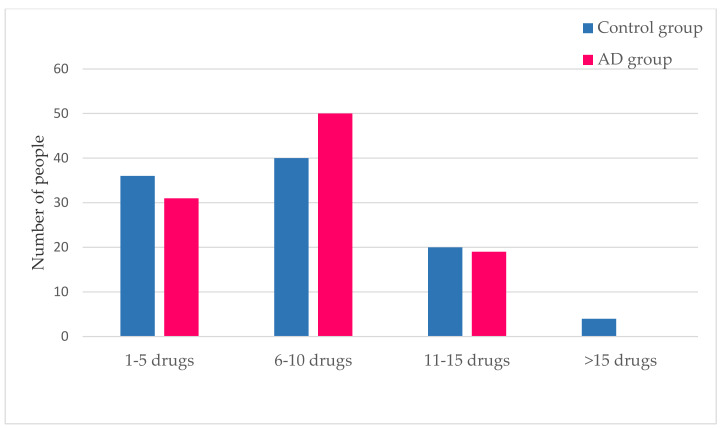
Number of medicines prescribed in the control and AD populations. This figure shows the amount of medication taken by subjects belonging to both the control and AD groups.

**Table 1 ijerph-17-03398-t001:** Presence of comorbidities with higher incidence in the AD group.

	Control Group	AD Group
Medical condition	%	%
Cataract	12.0%	21.0%
Urinary incontinence	16.0%	38.0%
Vitamin D deficiency	5.0%	11.0%
Osteoarthritis	9.0%	26.0%
Hypoacusis	1.0%	13.0%
Osteoporosis	1.0%	20.0%
Personality disorders	1.0%	12.0%

The percentage of individuals not affected by these comorbidities are the remaining quantities up to 100, since in each group there are a total of 100 individuals.

**Table 2 ijerph-17-03398-t002:** Chi-square data and significance values for cataracts, urinary incontinence, vitamin D deficiency, osteoarthritis, hypoacusis, osteoporosis, and personality disorders.

Medical Condition	Chi-Square	df	Sig.
Cataracts	2.940	1	0.086 *
Urinary Incontinence	12.278	1	0.000 *
Vitamin D deficiency	2.446	1	0.118
Degenerative joint disease	10.009	1	0.002 *
Hearing loss	11.060	1	0.001 *
Osteoporosis	19.207	1	0.000 *
Personality disorders	9.955	1	0.002 *

* The Chi-square statistic is significant at the level 1.

**Table 3 ijerph-17-03398-t003:** Presence of comorbidities with lower incidence in the AD group.

	Control Group	AD Group
Medical condition	**%**	**%**
Eye strain	9.0%	2.0%
Ictus	5.0%	1.0%
Vertigo	11.0%	2.0%
Hyperuricemia	14.0%	9.0%
Circulatory insufficiency	34.0%	21.0%
Atrial fibrillation	11.0%	7.0%
Respiratory failure	25.0%	4.0%

The percentage of individuals not affected by these comorbidities are the remaining numbers up to 100, since in each group there is a total of 100 individuals.

**Table 4 ijerph-17-03398-t004:** Chi-square data and significance values for eye strain, stroke, vertigo, hyperuricemia, circulatory failure, atrial fibrillation, and respiratory failure.

Medical Condition	Chi-Square	df	Sig.
Eye strain	4.714	1	0.030 *
Ictus	2.749	1	0.097 *^b^
Vertigo	6.664	1	0.010 *
Hyperuricemia	1.228	1	0.268
Circulatory insufficiency	4.238	1	0.040 *
Atrial fibrillation	0.977	1	0.323
Respiratory failure	17.786	1	0.000 *

* The Chi-square statistic is significant at the level 10. ^b^ More than 20% of the cells in this subtable had predicted cell counts less than 5. The Chi-square results may not be valid.

**Table 5 ijerph-17-03398-t005:** Presence of comorbidities that do not show significant differences in both groups.

	Control Group	AD Group
Medical condition	%	%
Type 2 diabetes mellitus	20.0%	19.0%
Glaucoma	6.0%	6.0%
Depression	26.0%	27.0%
Obesity	7.0%	7.0%
Arthritis	9.0%	8.0%
High blood pressure	64.0%	51.0%
Dyslipidemia	39.0%	45.0%
Anxiety	12.0%	14.0%
Heart disease	26.0%	31.0%

The percentage of individuals not affected by these comorbidities are the remaining numbers up to 100, since in each group there is a total of 100 individuals.

**Table 6 ijerph-17-03398-t006:** Chi-square data and significance values for type 2 diabetes mellitus, glaucoma, depression, obesity, arthritis, hypertension, dyslipidemia, anxiety, and heart disease.

Medical Condition	Chi-Square	df	Sig.
Type 2 diabetes mellitus	0.032	1	0.858
Glaucoma	0.000	1	1.000
Depression	0.026	1	0.873
Obesity	0.000	1	1.000
Arthritis	0.064	1	0.800
High blood pressure	3.458	1	0.063*
Dyslipidemia	0.739	1	0.390
Anxiety	0.177	1	0.674
Heart disease	0.613	1	0.434

* The Chi-square statistic is significant at the level, 10.

**Table 7 ijerph-17-03398-t007:** Comparative table of the action protocols of León University Hospital and Alzheimer’s Comprehensive Care Center of León.

	Comprehensive Care Center of León	León University Hospital
**Target**	Prevention and response in case of falls in patients with Alzheimer’s disease	Activities to be performed by the nursing staff according to the different nursing diagnosis lists of the NANDA that the individual with Alzheimer’s disease has
**Coincidences**	-Use of specific support measures for wandering, i.e., walking sticks or support rails -Specific protocol for physical restraint, if necessary, under medical order (with information to relatives) -Specific nursing care monitoring skin integrity or reassessment of the need for restraints -Increase environmental safety by avoiding slippery floors and architectural barriers, placing objects more easily accessible, positioning beds at a lower height and with handrails to prevent falls	
**The differences**	-Directed to all employees	-Directed only to the nursing staff
**Key points**	-The use of the J.H. Downton scale -It assesses the presence of certain intrinsic factors (age, drugs, or associated comorbidities) and extrinsic factors (of an environmental nature as inappropriate soil or equipment in each case) -There is an action protocol in case of falls -1st aid + 2nd reassures and secures the rest -Evaluate general condition and emergency equipment needs, report on the fall (questionnaire)	-Individualized protocol for the nursing staff
**Lacks**	-Lack of personalized care by the employees because there is a single protocol for all	-The J.H. Downton scale is not used -It does not take into account both intrinsic and extrinsic factors -There is no protocol for intervention in the event of a fall -Lack of protocols for the rest of the professional staff members working in the hospital

This table compares different points between the Protocol of the Alzheimer’s Center León and the Protocol of the University Hospital of León

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
