# Peer review of "Prevalence of Comorbidities in Individuals Diagnosed and Undiagnosed with Alzheimer’s Disease in León, Spain and a Proposal for Contingency Procedures to Follow in the Case of Emergencies Involving People with Alzheimer’s Disease"

_ijerph, 2020, doi:10.3390/ijerph17103398_

Round 1

Reviewer 1 Report

Comments to the authors:

The paper by Tortajada-Soler, M et al., presents some interesting findings, showing the prevalence of comorbidities associated (or not) to Alzheimer’s Disease (AD) in Leon Sapin population. Interestingly, the authors define in The Title and the Abstract sections the pathology description, the potential molecular mechanisms involved in this disease and, the risks and protectives factor associated with AD. In the methods section, there are some issues need to be clarified. Moreover, there is a  concern regarding the gender population study (female and male percent) and its relation to comorbidities, such as Osteoporosis, that need to be addressed before publication. Some of their results are different from previous reports. Finally, the discussion summarizes and compares their result against previously works from other investigators. However, the authors do not discuss the reason(s) or potential explanations of these discrepancies. 

Major comments

Material and Methods. 

Population study or Data collection. The authors need to provide the type of specialists (psychiatrist for example?) that carry out the criteria or protocols for this study.

Control group. 

The authors need to clarify whether the control group was evaluated with the same protocol or criteria for the AD group. There is another question, Was the selection of the subjects made randomly of this group?

The authors need to briefly describe all the methodology or studies that have been used for the diagnostic of comorbidities from tables (1, 2, 3). For example, the case of osteoporosis; briefly describes the methodology or clinical study used to determined osteoporosis in the participants. In the case of Vitamin D, what methodology the authors used, serum 25(OH)D measurement?.

Figures

Overall, the authors need to briefly describe the figures and be placed as footnotes below figures or tables.

Results and Discussion

As the authors (figure 1) and other investigators have been previously shown that the AD is more frequent in females than males. Since the AD group is around 70% of female, is there any possibility that some comorbidities are related to the nature of prevalent gender, and not necessarily to AD diseases. Specifically, it is been reported that osteoporosis affects women and men differently and is more frequent in females than males. Since AD group is around 70% of females, therefore is expected to have more prevalence of osteoporosis in the AD group. This explanation might be related to gender, but not to the AD. I will suggest to the authors present this data as figure 1, percent of comorbidities by gender, at least for osteoporosis case and discussed in the discussion section.

The authors need to discuss the potential (s) explanation of the differences of their results compare with previously results published by other researchers, such as 21 and 25. In previous publications (25 for example) the size of the groups are around thousand instead of hundreds. The author mentioned in the discussion would like to increase the size of the groups. However, is there any possibility that the size of the study affects the interpretation of the results? The author needs to discuss this question in the discussion section.

Reviewer 2 Report

The AD (Experimental) and Control groups are distinctly different in terms of age and gender. Line 208-209 "The difference is even more pronounced for osteoarthritis, which was present 209 in 26% of the individuals in the AD group compared with 9% in the control group." Since you have more women in the AD group (69%) than in the Control (53%) this would explain the higher rate of osteoarthritis an also comorbidities (Verbrugge, 1995). The AD group is markedly older that explains the increase in comorbidities.

Table 3 refers to physical complaints and conditions rather than morbidities that are defined as "the condition of being diseased." It would have been more readable to include only the Yes condition (with the No totals exactly 100% so not necessary). It is interesting that people with AD have fewer physical complaints. No information was provided on how the Comprehensive Care Center, León Alzheimer 155 Center, and the Alzheimer's Daycare Center of León gathered the information. This was likely (but unable to determine) though questionnaires and blood work.

Line 245 should read p-value of 0.05 (NOT p-value of 0.10) again line 247.

Table 5 delete No categories. The numerical decimal point is a full stop, not a comma.

Table 6 Alpha should be at 0.05, not 0.1, therefore there are no significant observed values in the chi-square results.

Table 7 is erratically formatted. Each row category should be in its own box.

Line 337 replace word "associated" with "due"

The conclusion that stroke does not result in AD cannot be made in a one-time study. Also, there was no retrospective assessment of the AD group to see if they experienced a stroke in the past. 

I think the title of the paper needs to be changed to reflect the content of the study. "Prevalence of comorbidities in individuals diagnosed with Alzheimer's disease compared to controls and fall response protocols, in León, Spain."

Overall Review:

The authors do not make the connection between the first part of their study (comorbidities) with the second part (protocols for falls). I am not sure there is a connection. As such this paper is two very different studies perhaps not worthy individually but the authors combined them together to enhance substance. It does not. The differences in age and gender between the experimental and control group diminish the results of the first part, while the lack of methodology for the second part addressing falls protocol diminish the second.

References

Verbrugge, L. M. (1995). Women, men, and osteoarthritis. Arthritis & Rheumatism: Official Journal of the American College of Rheumatology8(4), 212-220.
